

# Improved haplotype resolution of highly duplicated MHC genes in a long-read genome assembly using MiSeq amplicons

Samantha Mellinger[1], Martin Stervander[1,2,3], Max Lundberg[1], Anna Drews[1] and Helena Westerdahl[1]

[1] Department of Biology, Molecular Ecology and Evolution Lab, Lund University, Lund, Sweden
[2] Department of Biology and Environmental Science, Faculty of Health and Life Sciences, Linnaeus University, Kalmar, Sweden
[3] Bird Group, Natural History Museum, Tring, Hertfordshire, United Kingdom

Corresponding authors
Samantha Mellinger,
samantha.mellinger@biol.lu.se
Helena Westerdahl,
helena.westerdahl@biol.lu.se

## ABSTRACT

Long-read sequencing offers a great improvement in the assembly of complex genomic regions, such as the major histocompatibility complex (MHC) region, which can contain both tandemly duplicated MHC genes (paralogs) and high repeat content. The MHC genes have expanded in passerine birds, resulting in numerous MHC paralogs, with relatively high sequence similarity, making the assembly of the MHC region challenging even with long-read sequencing. In addition, MHC genes show rather high sequence divergence between alleles, making diploid-aware assemblers incorrectly classify haplotypes from the same locus as sequences originating from different genomic regions. Consequently, the number of MHC paralogs can easily be over- or underestimated in long-read assemblies. We therefore set out to verify the MHC diversity in an original and a haplotype-purged long-read assembly of one great reed warbler *Acrocephalus arundinaceus* individual (the focal individual) by using Illumina MiSeq amplicon sequencing. Single exons, representing MHC class I (MHC-I) and class IIB (MHC-IIB) alleles, were sequenced in the focal individual and mapped to the annotated MHC alleles in the original long-read genome assembly. Eighty-four percent of the annotated MHC-I alleles in the original long-read genome assembly were detected using 55% of the amplicon alleles and likewise, 78% of the annotated MHC-IIB alleles were detected using 61% of the amplicon alleles, indicating an incomplete annotation of MHC genes. In the haploid genome assembly, each MHC-IIB gene should be represented by one allele. The parental origin of the MHC-IIB amplicon alleles in the focal individual was determined by sequencing MHC-IIB in its parents. Two of five larger scaffolds, containing 6–19 MHC-IIB paralogs, had a maternal and paternal origin, respectively, as well as a high nucleotide similarity, which suggests that these scaffolds had been incorrectly assigned as belonging to different loci in the genome rather than as alternate haplotypes of the same locus. Therefore, the number of MHC-IIB paralogs was overestimated in the haploid genome assembly. Based on our findings we propose amplicon sequencing as a suitable complement to long-read sequencing for independent validation of the number of paralogs in general and for haplotype inference in multigene families in particular.

## INTRODUCTION

Multigene families contain paralogs (gene copies) that have evolved by repeated gene duplication and share high sequence similarity and similar functions (*Nei & Rooney, 2005*). Several multigene families have been identified in vertebrates, where they are involved for example in olfaction (*Niimura, 2012*), oxygen transport (*Hardison, 2012*) and immune functions (*Nei, Gu & Sitnikova, 1997*; *Alcaide & Edwards, 2011*). The major histocompatibility complex (MHC) is a genomic region that holds a wide range of immune-related genes, with the MHC class I (MHC-I) and MHC class II (MHC-II) genes being particularly noteworthy and well-studied (*Horton et al., 2004*; *Shiina et al., 2007*; *Shiina et al., 2009*; *Shiina et al., 2017*). The MHC molecules, encoded by the MHC genes, play a key role in antigen presentation and are essential for the initiation of adaptive immune responses (*Abbas, Lichtman & Pillai, 2020*).

In passerine birds (Aves: Passeriformes), the number of MHC paralogs have expanded massively leading to high MHC diversity (number of MHC genes or alleles per individual; *O'Connor et al., 2016*; *Minias et al., 2018*; *He, Minias & Dunn, 2021*). MHC paralogs are found in two genomic arrangements: as single copies or as tandemly duplicated copies. The latter category is represented by gene copies repeated over short intergenic distances and have been reported in a wide range of species (*Chen et al., 2015*; *Shiina & Blancher, 2019*; *Westerdahl et al., 2022*). Long-read sequencing facilitates the assembly of complex genomic regions (*van Dijk et al., 2018*), for example, regions with high repeat content and/or tandemly duplicated paralogs like MHC genes (*O'Connor et al., 2019*; *Vekemans et al., 2021*). MHC paralogs are often highly similar within species because of recent and repeated gene duplication or evolutionary homogenization due to gene conversion (*Wittzell et al., 1999*; *Goebel et al., 2017*; *Westerdahl et al., 2022*), and long reads may be of sufficient length to span across such genomic repeats. At the same time, there may be considerable sequence divergence between MHC alleles within the same gene (*Robinson et al., 2017*) and, at least in passerines, substantial gene copy number variation (CNV) between haplotypes. This was exemplified in a wild population of great reed warblers *Acrocephalus arundinaceus*, with 4–21 MHC-I alleles found per haplotype (*Roved et al., 2022*) across 559 individuals, which represents 2–11 MHC-I genes under full heterozygosity. On one hand, the large sequence difference between alleles within a gene facilitates the separation of haplotypes in a long-read assembly, though, on the other hand, such alleles may be incorrectly assigned as belonging to different loci in the genome rather than as alternate haplotypes from the same locus. Consequently, producing a correct haploid representation of the MHC region poses a challenge. Therefore, a haploid long-read sequencing genome assembly could either overestimate or underestimate the number of MHC paralogs. Hence, to fully appreciate the MHC genomic region in species with considerable CNV, it has been suggested that a pangenome rather than a single reference genome is preferable (*Vekemans et al., 2021*).

The genotyping of MHC in non-model organisms in ecological and evolutionary studies is nowadays most often performed using PCR-based amplification of focal exons that are then high-throughput sequenced (HTS). Thus, it is possible to simultaneously co-amplify all MHC alleles in every individual by PCR (*i.e.* amplicons) and then use HTS to genotype these PCR amplicons from large numbers of individuals at a low cost (*O'Connor et al., 2019*). Most studies of non-model organisms describe MHC diversity and MHC polymorphism (the number of different MHC alleles per paralog in a population) by targeting MHC-I exon 3 and MHC-IIB exon 2 (*Alcaide, Liu & Edwards, 2013*; *O'Connor et al., 2016*; *Biedrzycka et al., 2017*; *Minias et al., 2018*). These exons encode approximately 50% of the peptide-binding pocket of the MHC molecules, which present antigens and are highly polymorphic. Although PCR-based methods allow multilocus amplification (*Babik, 2010*), it is rare to be able to assign MHC alleles to specific MHC loci, particularly in species with high MHC diversity and CNV (*Vekemans et al., 2021*).

One way to circumvent the limitation of haploid representation of MHC based on long-read sequencing is to combine it with HTS amplicon sequencing and measure MHC diversity in a known genetic background, *i.e.,* in parents and offspring. Amplicon HTS data of family members combined with linkage analysis is a useful tool to infer haplotypes and the putative structure of the MHC genomic region. We recently characterized the MHC region in four passerine species (*Westerdahl et al., 2022*), based on long-read sequenced genomes, including a single great reed warbler individual (the focal individual; *Sigeman et al., 2021*). This great reed warbler had 15 full-length MHC-I and 56 full-length MHC-IIB genes in open reading frame, that were found on 16 different scaffolds of varying sizes (50–2,058 kbp). A large proportion of the MHC genes were organized as tandemly duplicated paralogs: 12 MHC-I genes and 49 MHC-IIB genes in open reading frame. The MHC region covered 5.5 Mbp, and included *e.g.*, additional MHC-II genes (MHC-IIA and MHC-DM), and MHC related genes, *i.e.,* genes expected to be found in the MHC region, such as the antigen peptide transporter genes, TAP1 and TAP2 (*Westerdahl et al., 2022*).

In the present study we genotyped the MHC-I and MHC-IIB diversity in the focal individual using amplicon HTS and evaluated the MHC-I and MHC-IIB diversity in the original long-read genome assembly (GRW Falcon-2017) and in the post-processed haploid genome assembly (Purge Haplotigs). In the Purge Haplotigs assembly alternate haplotypes had been removed to generate an improved haploid representation of the genome, though this procedure is challenging in scaffolds containing MHC genes due their repetitive nature and differences between haplotypes. Using amplicon HTS data from the parents of the focal individual, we overcame this challenge and determined the parental origin of the annotated MHC alleles, verifying the MHC haplotypes in the haploid long-read genome assembly.

## MATERIALS AND METHODS

### Genomic reference of the great reed warbler MHC region

The great reed warbler genome was characterized through a *de novo* genome assembly reconstructed from long-read sequencing (PacBio, Menlo Park, CA, USA), short-read

sequencing (Illumina, San Diego, CA, USA), and optical mapping data (BioNano, San Diego, CA, USA) from a female individual, our focal individual (*Sigeman et al., 2021*). The final genome assembly comprised 3,013 scaffolds, with a total size of 1.2 Gb of which half of the assembled genome was included in scaffolds that were at least 21.4 Mbp long (N50). As a first step, the genome was assembled using the Falcon assembler (v.0.4.2) (*Chin et al., 2013*), which separates sequences into primary contigs and associated contigs. Associated contigs represent alternative haplotypes and the primary contigs were used for subsequent scaffolding steps. As some primary contigs may represent incorrectly assigned associated contigs, which is more likely to happen in species with high heterozygosity, the Purge Haplotigs pipeline (*Roach, Schmidt & Borneman, 2018*) was used to remove sequences that were putative alternate haplotypes of other sequences based on read coverage and pairwise sequence similarities. Full-length MHC variants were identified in both versions of long-read assemblies, GRW Falcon-2017 and Purge Haplotigs, by blasting complementary DNA Sanger sequences, exons 2–4 as described in *Westerdahl et al. (2022)*, from eight MHC-I and seven MHC-IIB loci previously characterized from expressed messenger RNA in the focal individual (*Westerdahl, Wittzell & von Schantz, 1999*; *Westerdahl, Wittzell & von Schantz, 2000*). Note, MHC 'variants' are called alleles when we reference to the Falcon assembly and genes when we reference to the Purge Haplotigs.

## Amplicon sequencing

Amplicon sequencing was performed using high-throughput sequencing technology (Illumina MiSeq, San Diego, CA, USA) in order to genotype MHC-I and MHC-IIB genes in a great reed warbler family including the focal individual, two siblings as well as both parents. The birds are part of a long-term monitored wild population at Lake Kvismaren in Sweden (*Hansson et al., 2007*). The focal individual of this study was sacrificed in 1996 and all individuals in this study were blood sampled with the permission from the Swedish Environmental Protection Agency (Permit number M98-96). Genomic DNA (gDNA) was extracted using a modified phenol-chloroform DNA extraction method of blood samples (*Sambrook, Fritsch & Maniatis, 1989*). MHC-I loci were genotyped by PCR amplification of either a 246 bp or a 262 bp fragment within exon 3 with previously designed primers HNalla/HN46 (*Westerdahl et al., 2004a*; *Westerdahl et al., 2004b*) as described in *Roved et al. (2018)* and newly designed primers HNalla-1/R3Ex3b, respectively (Table S1). For MHC-IIB genotyping, four newly designed primer pair (PP) combinations (PP1, PP2, PP3, and PP5) were used to amplify MHC-IIB exon 2 and the fragments length obtained varied between 256 bp and 351 bp (Table S1).

## Filtering method and allele selection

For both MHC-I and MHC-IIB, only the amplicons sequenced from the focal individual and its parents were used in the present study. All three samples were run in replicates for all primer pair combinations except for HNalla/HN46. Obtained sequences were trimmed of the adapters and primer sequences using Cutadapt (*Martin, 2011*). Trimmed sequences were imported in RStudio v.1.2.1335 (*R Studio Core Team, 2020*) and filtered with the R packages DADA2 (*Callahan et al., 2016*) as described in *Stervander et al. (2020)* and *Drews*

*& Westerdahl (2019)* using the function filterAndTrim and adjusting settings for expected error rate and quality cut-off for each primer pair combination.

For the parents and the focal individual, the numbers of filtered reads for both MHC-I primer pairs (PPs) ranged between 19,193 and 34,310 (median 26,907). The maximum percentage of reads for an MHC-I amplicon allele (number of reads for a particular allele divided by the total number of reads for all alleles in an individual) was 10.8–12.9% and the minimum percentage of reads was 0.12–0.23% for all individuals. In addition, to keep an allele we took into account that (i) it should be found in both replicates of the same individual within PP combination or (ii) it should be found with both PPs in the same individual or (iii) it should be found in at least two individuals in the family independently of the PP combination (because not all alleles are expected to be amplified by all PPs). Following these rules, we retained two additional MHC-I alleles with low number of reads (0.06%), but which were amplified in the focal individual and its parents.

For MHC-IIB, we selected data from the two best-performing primer pair combinations (PPs) *i.e.,* for which we obtained the largest number of different MHC-IIB amplicon alleles post-trimming (PP1: 93 and PP5: 89). Only sequences of the expected length or that were shorter by a combination of a multiple of 3 bp (considering the length of a full codon) were kept. The two other MHC-IIB PPs gave a low number of unique amplicon alleles in the focal individual (PP2: 74 and PP3: 65) and data from both PPs were discarded (Table S1). The number of filtered reads for both PPs ranged between 29,960 and 51,190 (median 40,571). The maximum percentage of reads for an MHC-IIB amplicon allele was 2.9–5.8% and the minimum percentage of reads was 0.12–0.23% for all individuals. MHC-IIB genes are highly duplicated in the great reed warbler and many more paralogs are amplified compared to MHC-I (*Westerdahl et al., 2022*). Consequently, the read depth observed (*i.e.,* percentage of reads) for all MHC-IIB amplicon alleles is lower as many more alleles are amplified simultaneously in a single individual. In the focal individual, the number of unique MHC-I and MHC-IIB amplicon alleles amplified was assessed for each primer pair combination (Table S1). Finally, we identified identical amplicon sequences amplified by both PPs for MHC-I and for MHC-IIB using the online program Seqeqseq (http://130.235.244.92/apps/seqeqseq.html).

## Mapping of MHC-I and MHC-IIB HTS-amplicon alleles

In order to identify paternally and maternally inherited annotated MHC alleles, MHC-I and MHC-IIB amplicon alleles found in the focal individual were sorted into three categories. Amplicon alleles found in the focal individual and the father were identified as paternal (P) whereas amplicon alleles found in the focal individual and the mother were identified as maternal (M). Amplicon alleles found in the focal individual and both parents were identified as unresolved alleles (U).

For MHC-I, amplicon alleles amplified with both PPs represent a fragment of 246 or 262 bp within exon 3, which has a total length of 270 bp (Table S1). For MHC-IIB, the amplicon alleles amplified using PP5 represent a fragment of 256 bp within exon 2 whereas amplicon alleles amplified using PP1 were trimmed to correspond to the length of exon 2, which is 270 bp (Table S1). All MHC-I and MHC-IIB amplicon alleles found in the

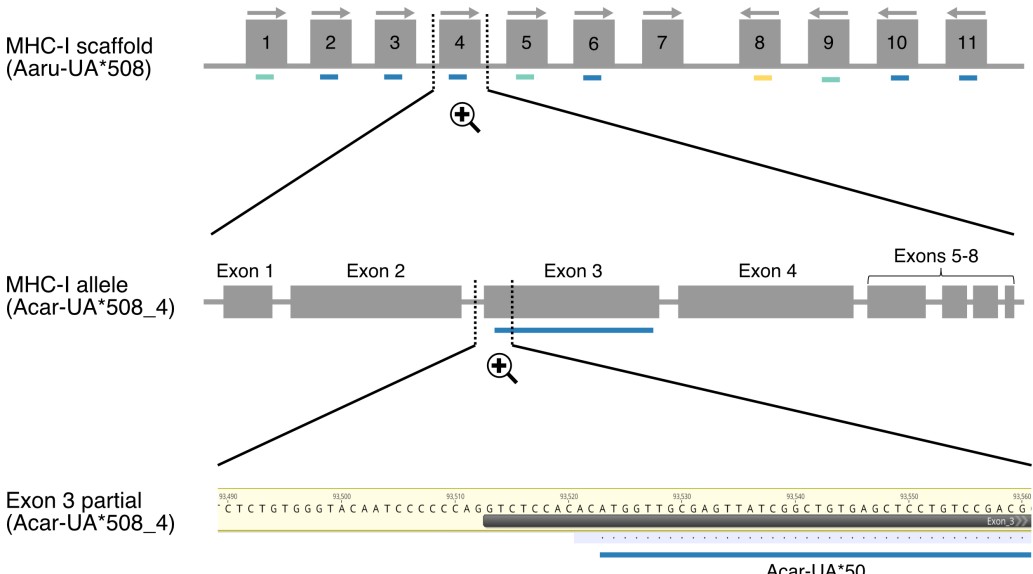

**Figure 1 Schematic illustration showing the mapping of amplicon alleles (HTS) to an annotated scaffold in the Falcon-2017 assembly of the focal individual.** Annotated MHC-I alleles (grey boxes) are numbered based on their position on the scaffold Aaru_508 and allele orientations are indicated with arrows. Amplicon alleles that have mapped to the exon 3 sequence of annotated MHC-I alleles (detected MHC-I alleles) are indicated with colored dashes (blue for paternal alleles, yellow for maternal alleles and turquoise for unresolved alleles). Each amplicon allele that mapped to an annotated allele was recorded, exemplified here with amplicon allele "Acar-UA*50" (referred as P-7 in Table S3) that mapped to the annotated MHC-I allele Acar-UA*508_4.

focal individual were mapped using Geneious Prime® 2020.1.2 (*v.* 11.0.6) to the original great reed warbler long-read genome assembly (GRW Falcon-2017) and to post-processed haploid genome assembly (Purge Haplotigs) which has gone through the post-assembly filtering procedure (Fig. 1, Tables S2–S4). We first used the Geneious RNA mapper with the following settings: Custom Sensitivity and allowing 0% mismatches per read. We also performed a complementary mapping step with more relaxed settings using the Geneious mapper (custom sensitivity, allowing 1–4% mismatches per read). Allowing mismatches can help to identify putative alternate (second) alleles for a gene copy if the alleles only differ by one or two nucleotides. It is also a way to account for remaining sequencing errors or indels in genome assemblies during the mapping step. Finally, nucleotide sequence similarities (based on pairwise similarity matrices) were compared between amplicon alleles and the corresponding coding exon sequences of annotated MHC-I and MHC-IIB alleles in both long-read genome assemblies.

Amplicon alleles were assigned to annotated MHC alleles when they shared identical nucleotide sequences (Fig. 1, Tables S3–S4). Thus, an amplicon allele could be assigned to multiple annotated MHC alleles if the coding sequence was identical with each of them (based on nucleotide sequence of exon 3 for MHC-I alleles and exon 2 for MHC-IIB alleles). Additional verifications were performed to confirm the assignment of amplicon alleles that were not identical to any annotated MHC alleles based on nucleotide sequence similarity.

Two different conditions were tested: (1) to be assigned, a single amplicon allele should have at least 99% nucleotide sequence similarity (corresponding to one or two nucleotide substitutions) with a unique annotated MHC allele; (2) to be considered as two allelic variants of a unique annotated MHC gene, two amplicon alleles should map and share high nucleotide sequence similarity (>99%) with each other and with a unique annotated MHC allele. Amplicon alleles mapping with lower nucleotide sequence similarity were not considered.

## Coding sequence similarity of MHC-IIB haplotypes

In the great reed warbler, the MHC-IIB genes have expanded rather recently and most MHC-IIB gene copies are similar to each other (*Westerdahl et al., 2022*). However, MHC-IIB alleles originating from a single gene copy are expected to be more similar than alleles originating from different gene copies. Thus, we expect MHC-IIB alleles from two complementary haplotypes to be more similar to each other than MHC-IIB alleles from different genomic regions, as measured by pairwise sequence distances. We extracted all 56 full-length MHC-IIB genes in open reading frame (ORF, putatively functional) found in the haploid representation of the great reed warbler genome assembly Purge Haplotigs and aligned the sequences using the Geneious alignment in Geneious Prime® 2021.1.1. We computed between-group mean distances by calculating nucleotide pairwise distance between all MHC-IIB gene copies in scaffolds which had >5 tandemly duplicated genes (scaffolds: Aaru-DAB*554, Aaru-DAB*357, Aaru-DAB*120, Aaru-DAB*301 and Aaru-DAB*45) in MEGA version 11 (*Tamura, Stecher & Kumar, 2021*), considering pairwise deletion and performing 1,000 bootstraps replicates. The same analyses were performed for full-length amino acids sequences. *Westerdahl et al. (2022)* suggested that the two scaffolds Aaru-DAB*554 (maternally inherited) and Aaru-DAB*357 (paternally inherited) may represent the two parental haplotypes of a single MHC-IIB genomic region, *i.e.,* the two plausible complementary haplotypes. To investigate this further we computed the mean nucleotide pairwise distance between each annotated MHC-IIB gene belonging to scaffold Aaru-DAB*554 and each of the MHC-IIB genes at the four scaffolds with >5 tandemly duplicated genes. The same analysis was also performed for Aaru-DAB*357.

# RESULTS

## MHC diversity in the long-read genome assemblies

In the original long-read genome assembly of the focal individual, GRW Falcon-2017, 25 MHC-I and 96 MHC-IIB full-length alleles were annotated in primary contigs (four MHC-IIB alleles were found in associated contigs), hence the primary contigs of GRW Falcon-2017 represent a diploid version of the MHC region. Following the post-assembly procedure (Purge Haplotigs), which aimed to remove sequences that represent alternative haplotypes of other sequences and give a haploid representation of the genome, 18 MHC-I and 66 MHC-IIB full-length genes were included and of these, 15 MHC-I and 56 MHC-IIB genes contained a predicted full-length open reading frame (ORF, *Westerdahl et al., 2022*). In total, 16 scaffolds with 1–19 MHC genes were identified in the great reed warbler Purge Haplotigs assembly (three MHC-I scaffolds, 11 MHC-IIB scaffolds and two scaffolds with

**Table 1   MHC diversity in the focal individual based on amplicon HTS and long-read genome assemblies.** MHC-I and MHC-IIB allelic diversity in the focal individual found with amplicon HTS (number of amplified alleles for MHC-I exon 3 and MHC-IIB exon 2) and full-length annotated MHC alleles in the two genome assemblies: the Falcon-2017 assembly, which for the MHC region contains both primary contigs and alternative haplotypes of the primary contigs, and the Purge Haplotigs assembly, for which most alternative haplotypes have been removed. The number of MHC alleles that contained an open reading frame are stated in brackets.

|  | MHC-I | MHC-IIB |
|---|---|---|
| Amplicon alleles (HTS) | 29 (22) | 95 (85) |
| Annotated alleles in the Falcon-2017 assembly | 25 (18) | 100 (87) |
| Annotated alleles in the Purge Haplotigs assembly | 18 (15) | 66 (56) |

both MHC-I and MHC-IIB genes). Of the 16 MHC scaffolds, five scaffolds with MHC-IIB genes were larger (103, 178, 175, 510 and 755 kbp) and had 6–13 tandemly duplicated MHC-IIB genes, *i.e.,* an MHC gene with at least one MHC gene as nearest neighbor within a short intergenic distance (on average 6,443 bp between tandemly duplicated MHC-IIB genes).

## MHC diversity based on amplicon HTS

Twenty-nine MHC-I alleles and 95 MHC-IIB alleles were amplified in the focal individual using amplicon HTS (22 MHC-I exon 3 alleles and 85 MHC-IIB exon 2 alleles contained an ORF; Table 1). The primers for amplicon sequencing were carefully designed to amplify the majority of all available MHC-I and MHC-IIB alleles in the great reed warbler (Table S1). Twelve out of 29 MHC-I amplicon alleles amplified in the focal individual were identified as paternal alleles (P; blue), seven were identified as maternal alleles (M; yellow) and nine were unresolved (U; turquoise; Fig. 2A). For MHC-IIB, 36 out of 95 amplicon alleles in the focal individual were paternal, 34 alleles were maternal, and 25 alleles were unresolved (Fig. 2A).

## MHC diversity in amplicons compared to MHC diversity in the GRW Falcon-2017 assembly

For MHC-I, 84% (21 alleles of the 25 full-length alleles) of the annotated alleles in GRW Falcon-2017 were successfully detected by amplicon alleles in the focal individual (Fig. 3, Fig. S1, Table S3). For MHC-IIB, the success rate was slightly lower, 78% (78 of the 100 full-length alleles (96 on primary and four on associated contigs)) of the annotated MHC alleles in GRW Falcon-2017 were detected using amplicon sequencing (Fig. 3, Fig. S1, Table S4). The total MHC diversity in amplicons, 29 MHC-I and 95 MHC-IIB alleles, was comparable to the diversity of the annotated MHC alleles in GRW Falcon-2017 assembly (25 MHC-I and 100 MHC-IIB, including associated contigs; Table 1, Fig. S1).

Sixteen MHC-I exon 3 amplicon alleles mapped to 21 annotated MHC-I genes in GRW Falcon-2017 (Fig. 3, Tables S2–S3). Four of these 16 MHC-I amplicon alleles mapped multiple times to a total of nine annotated MHC-I alleles (one amplicon allele mapped to three annotated alleles and three amplicon alleles mapped to two annotated alleles each; Fig. S1), and therefore most of the annotated alleles in GRW Falcon-2017 (84%) were detected. However, 13 of 29 MHC-I amplicon alleles (45%) did not map to any full-length

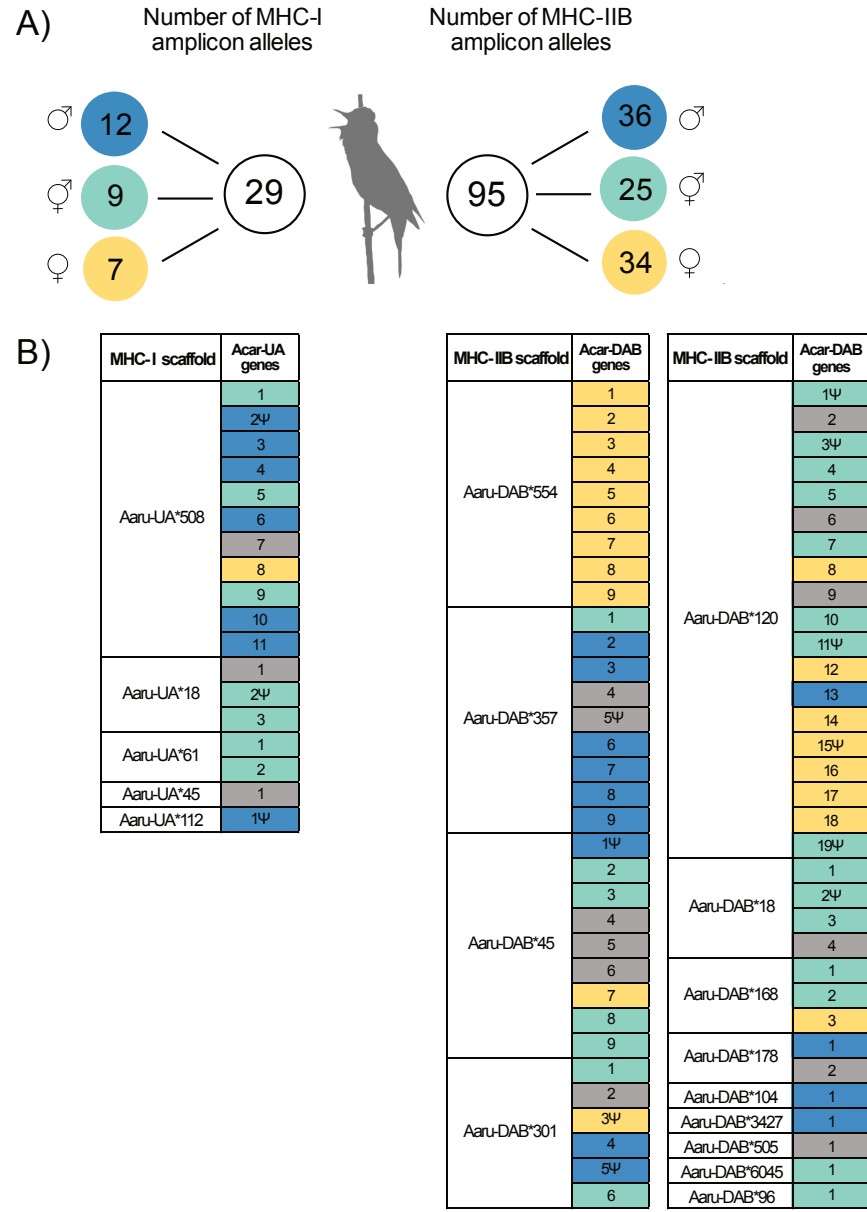

**Figure 2 Amplicon alleles (HTS) and annotated genes (Purge Haplotigs assembly) of the focal individual with the putative allelic parental origin indicated.** (A) Number of MHC amplicon alleles and their parental origin in the focal individual. Amplicon alleles were separated into three categories based on their inheritance in the focal individual: paternal alleles (blue), maternal alleles (yellow) and unresolved alleles (turquoise). Note that one MHC-I amplicon allele was found in the focal individual but was not successfully amplified in its parents. (B) Inferred parental origin for annotated MHC genes in the Purge Haplotigs assembly using the MHC amplicon allele information. MHC scaffolds are indicated as "Aaru-UA*" for MHC-I genes and "Aaru-DAB*" for MHC-IIB genes. Gene copies are indicated as "Acar-UA" and "Acar-DAB" and named after their position on annotated primary scaffolds in the Purge Haplotigs assembly. Non-functional genes are indicated with the symbol Ψ. Annotated MHC genes with no matching or no assigned amplicon alleles are in grey (see Methods).

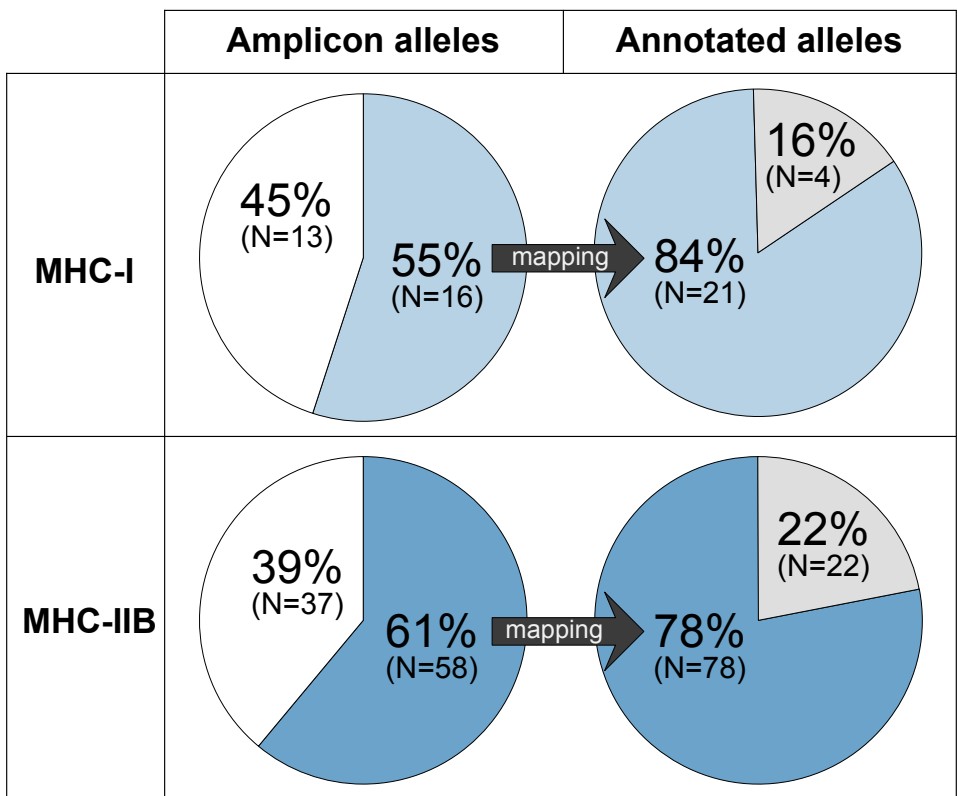

**Figure 3** **A moderate proportion of the MHC-I and MHC-IIB amplicon alleles (HTS) detects a considerable proportion of the annotated alleles (Falcon-2017 assembly).** Amplicon alleles mapping to annotated alleles (detected alleles) are indicated in blue (MHC-I: upper panel, light blue; MHC-IIB: lower panel, dark blue), amplicon alleles that were not mapping are indicated in white, and annotated alleles with no matching amplicon alleles are indicated in grey.

annotated MHC-I allele, suggesting that the genome assembly lacks annotations for many MHC-I alleles and/or that several MHC-I exon 3 amplicon alleles are PCR products from gene fragments. The latter is supported by two of the 13 MHC-I amplicons, which map to locations without full-length annotated MHC-I alleles in the GRW Falcon-2017 assembly (Fig. S1).

Fifty-eight MHC-IIB exon 2 amplicon alleles mapped to 78 annotated full-length MHC-IIB alleles in GRW Falcon-2017 (Fig. 3, Table S4). Ten of these amplicon alleles mapped multiple times to a total of 32 annotated MHC-IIB alleles that shared identical nucleotide sequences for exon 2 but differed elsewhere in the gene sequence (Fig. S1). However, 37 of 95 MHC-IIB amplicon alleles (39%) did not map to any full-length annotated MHC-IIB allele, suggesting an incomplete genome assembly and/or that MHC-IIB exon 2 amplicons are PCR-products from gene fragments. Again, the latter scenario is supported by seven of the 37 amplicons, which map to locations without full-length annotated MHC-IIB alleles in the GRW Falcon-2017 assembly (Fig. S1).

## MHC diversity in amplicons compared to MHC diversity in the Purge Haplotigs assembly

For MHC-I, 89% (16 of the 18 full-length genes) of the annotated MHC genes in Purge Haplotigs were successfully detected by amplicon alleles in the focal individual (Fig. 2B, Table S3). The success rate was again slightly lower for MHC-IIB, where 82% of the annotated MHC genes in Purge Haplotigs (54 of the 66 full-length genes) were detected using amplicon sequencing (Fig. 2B, Table S4). Thirteen MHC-I amplicon alleles mapped perfectly to 16 annotated full-length MHC-I genes (Acar-UA genes, Fig. 2B), of which two MHC-I amplicon alleles mapped multiple times (Table S3; note that one amplicon allele in the focal individual was not found in either parent, see Tables S2–S3). Fifty-three MHC-IIB exon 2 amplicon alleles mapped to 54 annotated MHC-IIB genes (Acar-DAB genes, Fig. 2B, Table S4) of which three MHC-IIB amplicon alleles mapped multiple times.

## Haplotype sorting of tandemly duplicated MHC genes in the Purge Haplotigs assembly

The parental amplicon alleles that mapped to the annotated full-length tandemly duplicated MHC-I and MHC-IIB genes in the Purge Haplotigs assembly show which scaffolds that are dominated by paternal (P) and maternal (M) MHC genes. Thus, since seven MHC-I genes were assigned as paternal and only one was assigned as maternal on scaffold 508 (Acar-UA*508), this scaffold is likely to represent a paternal haplotype (Fig. 2B, Table S3).

There were 54 MHC-IIB genes in the Purge Haplotigs assembly with information about parental origin (13P, 19M, and 22U, Fig. 2B, Table S4), and among the five larger scaffolds (103–755 kbp) with 6–13 tandemly duplicated MHC-IIB genes, one scaffold was putatively paternal (Acar-DAB*357), two were putatively maternal (Acar-DAB*554 and *120) and two were mixed (Acar-DAB*45 and *301).

One way of assessing whether the number of MHC-IIB genes is overestimated in the Purge Haplotigs assembly is to investigate whether any of the paternal and maternal scaffolds with tandemly duplicated genes are likely to represent haplotypes of the same genomic region. The nucleotide sequence of the MHC-IIB alleles is expected to be more similar within than between paralogs, so the average nucleotide distance between alleles from complementary haplotypes should be lower than between non-complementary haplotypes. We thus compared the average nucleotide pairwise distances (p-distances) of the tandemly duplicated MHC-IIB genes containing an ORF between the five larger scaffolds to investigate whether any pair-wise comparison among the scaffolds represent putatively complementary haplotypes. The mean nucleotide p-distance between genes in the two scaffolds Aaru-DAB*554 and Aaru-DAB*357 was lower (0.057) than the mean nucleotide p-distances of other scaffold comparisons (p-distance: 0.077–0.084; Fig. 4A). The same between scaffold difference was observed for the mean amino-acid pairwise distance (Aaru-DAB*554 and Aaru-DAB*357 p-distance: 0.093; other scaffolds p-distance: 0.123–0.134; Table S3).

Then we set out to compare the allelic distances on the two scaffolds Aaru-DAB*554 and Aaru-DAB*357 in more detail. First, we compared the nucleotide p-distance between each annotated gene on scaffold Aaru-DAB*357 and each gene on the other four larger
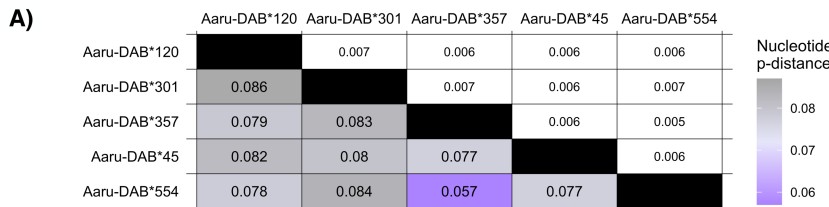

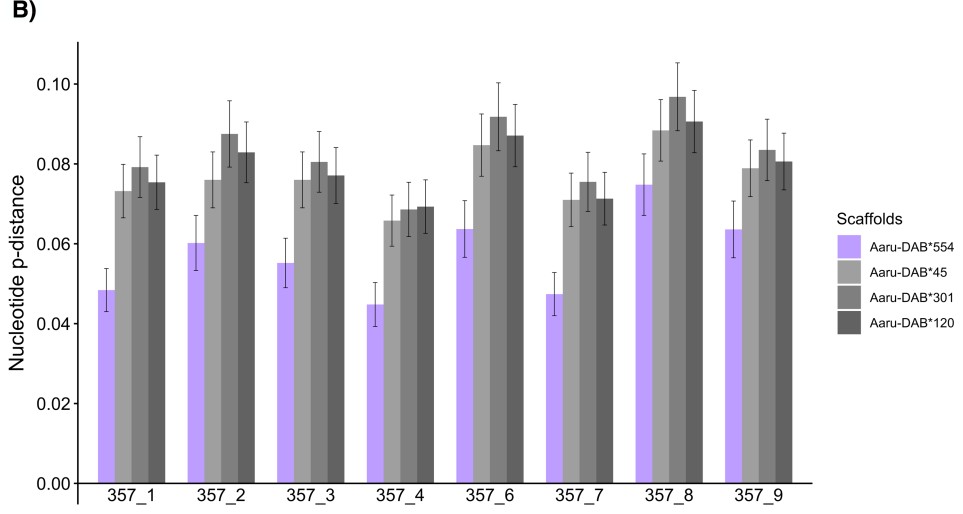

**Figure 4** **Mean nucleotide pairwise distances (p-distances) between tandemly duplicated MHC-IIB gene copies contained in an open reading frame in five scaffolds with >5 tandemly duplicated genes (Purge Haplotigs assembly).** (A) Heatmap of between-scaffold mean *p*-distances (calculated as the mean of all pairwise comparison between MHC-IIB gene copies contained in an open reading frame at two scaffolds (Aaru-DAB*), below diagonal) and standard errors (above diagonal). (B) Mean p-distances computed between each MHC-IIB gene copy from scaffold Aaru-DAB*357 (Acar-DAB*357_1–4;6–9) and all MHC-IIB gene copies at four scaffolds: Aaru-DAB*554 (purple), Aaru-DAB*45 (light grey), Aaru-DAB*301 (medium grey) and Aaru-DAB*120 (dark grey).

scaffolds. The nucleotide distance per gene was always smaller when compared to genes on scaffold Aaru-DAB*554 than compared with genes on the three other scaffolds (Fig. 4B). Second, we compared the nucleotide p-distance between each annotated gene on scaffold Aaru-DAB*554 and each gene on the four larger scaffolds. Likewise, the mean nucleotide p-distance between each annotated gene on scaffold Aaru-DAB*554 was smaller when compared to genes on scaffold Aaru-DAB*357 than compared with genes on the three other scaffolds (Fig. S2). Both the average p-distance of all genes per scaffold and the p-distance per gene are smaller between scaffolds Aaru-DAB*554 and Aaru-DAB*357 compared with the other larger scaffolds, suggesting that scaffolds Aaru-DAB*554 and Aaru-DAB*357 represent both haplotypes of the same genomic region.

## DISCUSSION

The MHC allelic diversity found for both MHC-I and MHC-IIB in the original long-read sequencing genome assembly (GRW Falcon-2017) was largely recovered using amplicon HTS data (84% of the annotated MHC-I alleles and 78% of MHC-IIB). Amplicon alleles (MHC-I exon 3 and MHC-IIB exon 2) frequently mapped to more than one full-length annotated MHC allele in the genome assembly, highlighting the fact that the MHC diversity based on amplicon alleles to some extent underestimates the true MHC diversity. This is because the amplicon alleles only represent a portion of the full-length annotated alleles of MHC genes in the genome and several annotated full-length MHC genes share identical MHC-I exon 3 and MHC-IIB exon 2 sequences in the great reed warbler. However, we think that MHC diversity based on amplicon alleles also may overestimate the true MHC diversity of full-length genes. This is because short amplicons sequences (MHC-I exon 3 and MHC-IIB exon 2) at times are derived from remnants of full-length genes, *i.e.,* pseudogenes where MHC-I exon 3 and MHC-IIB exon 2 contain an open reading frame.

The largest part of the MHC diversity, *i.e.,* annotated full-length alleles, described in the GRW Falcon-2017 assembly was detected using only approximately half of the MHC-I and MHC-IIB amplicon alleles (55% for MHC-I and 61% for MHC-IIB). This was a little unexpected, given that the GRW Falcon-2017 assembly reflects the diploid representation of the MHC region and is a rather high-quality assembly where the MHC scaffolds have been carefully annotated. Therefore, it suggests that the genome assembly does not describe the full MHC diversity. Nevertheless, one additional explanation for the low MHC diversity in the genome assembly compared to the MHC diversity based on amplicon alleles is linked to how MHC genes evolve. The MHC multigene family is believed to evolve according to the "birth and death" model (*Nei, Gu & Sitnikova, 1997*; *Burri et al., 2010*): novel MHC gene copies arise by gene duplication and some gene copies are maintained as functional genes whereas other genes become pseudogenes or are lost entirely. In our study, a small proportion (approximatively 7%) of the MHC-I and MHC-IIB amplicon alleles mapped with high support to genomic locations without annotated MHC genes. These genomic locations likely hold remnants of MHC alleles that happened to be amplified by our MHC-I and MHC-IIB amplicon primers. Finally, it cannot be excluded that some amplicon alleles are artefacts (*Babik, 2010*), although we expect artefact alleles to be very rare, as the majority of amplicon alleles were found in at least two individuals, *i.e.,* in different genetic backgrounds where the likelihood of identical PCR artefacts is expected to be low.

Using the primary contigs from the original long-read assembly and a subsequent removal of incorrectly assigned alternate contigs in Purge Haplotigs, we expected to obtain a haploid representation of the MHC region (*Roach, Schmidt & Borneman, 2018*). However, the correct assignment of haplotypes is challenging to perform in complex genomic regions like the MHC region in passerines (*Vekemans et al., 2021*). Tandemly duplicated MHC-I and MHC-IIB genes in passerines are found in repeat-rich regions and can evolve by simultaneous duplications of several MHC gene copies, resulting in genomic MHC regions with high sequence similarity that are demanding to assemble correctly. *Westerdahl et al. (2022)* inferred the gene duplication history of the tandemly duplicated

MHC-I genes in scaffold Aaru-UA*508 and of the tandemly duplicated MHC-IIB genes in scaffold Aaru-DAB*120 based on CDS similarity, gene size and intergenic distance, and suggested that three MHC gene copies had been duplicated on each scaffold.

In the present study we are less interested in the CDS similarity within scaffolds and more interested in the CDS similarity between scaffolds. Westerdahl et al. (2022), using the data from the present study, noted the presence of an MHC-IIB scaffold with entirely maternal alleles (Aaru-DAB*554) and another with mainly paternal alleles (Aaru-DAB*357), and suggested that the Purge Haplotigs analysis failed to recognize at least some sequences as complementary haplotypes. Here, we expand the characterization of scaffolds Aaru-DAB*357 and Aaru-DAB*554 by calculating genetic distances between the five largest MHC-IIB scaffolds with tandemly duplicated genes, revealing lower nucleotide distance between the maternal and paternal scaffold than in other between-scaffold comparisons. This finding lends further support for Aaru-DAB*554 and Aaru-DAB*357 being alternate haplotypes of the same genomic region, and it is likely that only one of the two scaffolds should be kept in the haploid representation of the MHC region assembly.

The MHC-II molecule is formed by one alpha and one beta chain. In the great reed warbler genome assembly, there are large numbers of MHC-IIB paralogs, which encode the beta chain, though the assembly only contains a single MHC-IIA gene, which encodes the alpha chain (Westerdahl et al., 2022). The single MHC-IIA gene is found next to the MHC-IIB paralogs on scaffold Aaru-DAB*554. It is likely that each MHC-IIB paralog on scaffold Aaru-DAB*554 can form an MHC-II molecule encoded by the single MHC-IIA gene, as can each MHC-IIB paralog on scaffold Aaru-DAB*357. Therefore, with the limited MHC-IIA diversity, diversifying selection of antigen-presenting MHC-II molecules seem to have favored repeated expansion of MHC-IIB genes through tandem duplications, resulting in both high diversity and high divergence among the MHC-IIB paralogs. However, as shown in the present study, the MHC-IIB diversity in the great reed warbler have been slightly exaggerated and six of the 56 MHC-IIB genes should be subtracted from Purge Haplotigs assembly. Interestingly, the zebra finch *Taeniopygia guttata*, another passerine bird, has 15 MHC-IIB genes next to its single MHC-IIA gene, and we cannot exclude that the great reed warbler has more than the nine annotated MHC-IIB genes next to its single MHC-IIA gene.

The assembler resolved 11 MHC-I paralogs organized in tandem on scaffold Aaru-UA*508 in Purge Haplotigs. Most of the annotated MHC-I genes were paternal, and only one annotated MHC-I gene (Acar-UA*508_8) was maternal. Likewise, there were more paternal ($N = 12$) than maternal ($N = 7$) MHC-I amplicon alleles in the focal individual, but we find it unlikely that the focal individual only would inherit a single maternal MHC-I allele that contained an ORF as indicated in the GRW Falcon-2017 assembly. Hence, we believe that some of the maternal MHC-I genes failed to be assembled. We envision that the assembly procedure has mixed maternal and paternal haplotypes in the genome interval, and that the use of future long-read sequencing approaches (*e.g.,* High Fidelity reads) will be able to yield a more correct haplotype assignment and gene order.

The reliability of the mapping of amplicon alleles to the annotated full-length alleles in both genome assemblies was high. Less stringent thresholds for mapping were tested

but did not improve the mapping efficiency (Table S2). We set the threshold of similarity between amplicon alleles and annotated full-length alleles to 100% for MHC-I and 99% for MHC-IIB, *i.e.,* two nucleotide differences in the latter sequence comparisons. Multiple primer pairs were designed to amplify the full MHC diversity. However, four annotated full-length MHC-I alleles and 22 MHC-IIB alleles were not detected by any amplicon alleles during the mapping. One reason can be that none, or only one of the amplicon primers, among all primer pairs tested, annealed satisfactorily and therefore these alleles were never amplified. When mapping the PCR primers to the genome such failure of PCR amplification potentially explains why two MHC-I and five MHC-IIB annotated full-length MHC alleles, respectively, were not detected.

## CONCLUSION

Most amplicon MHC alleles could be mapped to full-length annotated MHC alleles in the long-read assembly and our study shows that combining long-read sequencing and amplicon HTS is a valuable method to verify genetic diversity in multigene-families such as MHC. Parental amplicon alleles are useful for identifying maternal and paternal origins of annotated MHC alleles and also to confirm alternate haplotypes in the genome assembly. Calculating pairwise genetic distances between tandemly duplicated genes on different scaffolds is another useful complement to the post-assembly procedure to identify alternate haplotypes.

## ACKNOWLEDGEMENTS

We would like to thank Hanna Sigeman for her help in providing additional information on the great reed warbler genome assembly used in this study.

### Funding
This work was supported by the European Research Council (ERC) under the European Union's Horizon 2020 Research and Innovation Programme (grant number 679799 to Helena Westerdahl), the Swedish Research Council (grant numbers 2015-05149, 2020-04285 to Helena Westerdahl) and by the Jörgen Lindström's Foundation (grant number 137301 attributed to Samantha Mellinger). The funders had no role in study design, data collection and analysis, decision to publish, or preparation of the manuscript.

### Grant Disclosures
The following grant information was disclosed by the authors:
European Research Council: 679799.
Swedish Research Council: 2015-05149, 2020-04285.
Jörgen Lindström's Foundation: 137301.

### Competing Interests
The authors declare there are no competing interests.

## Author Contributions

- Samantha Mellinger conceived and designed the experiments, performed the experiments, analyzed the data, prepared figures and/or tables, authored or reviewed drafts of the article, and approved the final draft.
- Martin Stervander performed the experiments, authored or reviewed drafts of the article, and approved the final draft.
- Max Lundberg analyzed the data, authored or reviewed drafts of the article, and approved the final draft.
- Anna Drews performed the experiments, authored or reviewed drafts of the article, and approved the final draft.
- Helena Westerdahl conceived and designed the experiments, authored or reviewed drafts of the article, and approved the final draft.

## Animal Ethics

The following information was supplied relating to ethical approvals (i.e., approving body and any reference numbers):

The focal individual of this study was sacrificed in 1996 and all individuals in this study were blood sampled with the permission from the Swedish Environmental Protection Agency.

## DNA Deposition

The following information was supplied regarding the deposition of DNA sequences:

17 sequences of the same length (263 bp) are available at GenBank: MH468838; MH468849; MH468857; MH468874; MH468954; MH468960; MH468992; MH469005; MH469010; MH469033; MH469055; MH469097; MH469099; MH469109; MH469111; MH469115; MH469150.

The remaining 12 amplicon sequences for MHC-I and 95 amplicon sequences for MHC-IIB used in the mapping procedure are available in the Supplementary Files and GenBank: OP524014 to OP524120.

## Data Availability

The great reed warbler genome assembly acrAru1 is available at BioProject: PRJNA765537 *Sigeman et al., 2021*

The MHC contigs are available at Dryad along with manually curated fasta-files of full-length MHC-I and MHC-IIB genes in open reading frame: Westerdahl, Helena (2022), The genomic architecture of the passerine MHC region: high repeat content and contrasting evolutionary histories of single copy and tandemly duplicated MHC genes, Dryad, Dataset, https://doi.org/10.5061/dryad.fqz612jv6.

The raw sequencing data from amplicons (Illumina MiSeq) are available at NCBI BioProject: PRJNA913109.

## Supplemental Information

Supplemental information for this article can be found online at http://dx.doi.org/10.7717/peerj.15480#supplemental-information.

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
