# Peer review of "Improved haplotype resolution of highly duplicated MHC genes in a long-read genome assembly using MiSeq amplicons"

_PeerJ, doi:10.7717/peerj.15480_

## Round 0.1 · original submission · Minor Revisions

Dear authors,

After reading the reviews and comments, one reviewer accepted the article and the other reviewer suggested minor corrections, however, the first reviewer has some concerns that the study presents original findings, but their breadth is limited- all the conclusions are reached based on (in-depth) study of one individual case, likewise makes some suggestions about the experimental design and suggests that English should be reviewed by a fluent speaker. for this reason, he/she suggests that minor corrections be carried out before it can be accepted for publication.

Sincerely,

Armando Sunny

Reviewer 1 ·

Basic reporting

The article is written in professional and clear English. Introduction provides background, literature seems well referenced and relevant. However, the authors should provide more context as to how this work relates to the one shown in Westerdahl et al. 2022 (see also notes in the next section). The general structuring of the MHC region (as found in Westerdahl et al. 2022) should be better described in the introduction, because as is, the reason for focusing on certain scaffolds is unclear – e.g., a summarizing figure (or table) with information on how many MHC-conating contigs/scaffolds were analyzed, their names, whether something else was there e.g., MHC related/non-related genes; use of the term “core MHC” as provided in Westerdahl et al. 2022). While evidently this work is complementary to that of Westerdahl et al. 2022, this must be a stand-alone article that can be well understood without the need to read Westerdahl et al. 2022.

Structure is appropriate, with minor stipulations provided below. Discussion in well written and provides a good summary of the findings.

Figures are of good quality, and expect for figure 3 - easy to understand. Figure 3 could use some redesigning to be more readily understandable. Supplementary tables and figures lack titles and captions – these should be added.

There is no statement as to where the raw data is supplied – this should be provided.

Experimental design

The manuscript presents original primary research, by and large within the scope of the journal.

Nonetheless, while the research question is well defined and timely, it is not easy to identify the gap without knowledge of what was already shown in Westerdahl et al. 2022. In fact, it seems that this manuscript, which is currently under review, has already been referenced in the Westerdahl et al. 2022 – as “Mellinger et al. Manuscript” (but without any further information, e.g., Preprint server doi where it might have been deposited). I would imagine that this manuscript have been in preparation (or under revision somewhere else then PeerJ), when Westerdahl et al. 2022 was being published, but since there is no proper citation of this work - Westerdahl et al. 2022 was, in fact, the place where some of the findings of this manuscript were published for the first time…

Validity of the findings

Conclusions are well stated and linked to original research question. I suppose detailed information on underlying data is in Westerdahl et al. 2022 and some of it is in the supplementary tables (i.e., allele Genbank IDs) but it should be explicitly provided as a data availability statement – please add it.

Additional comments

Overall, presented work does provide useful insight and could be interesting for researchers trying to gauge the capacity to reconstruct complex genomic regions, such as MHC, using current genome assembly methods. It also compares results obtained with more “standard”, amplicon-based methods of MHC genotyping in no-model species with loci detected in said assembled genomes. Moreover, it brings valuable information on how both approaches can simultaneously underestimate and overestimate diversity, yet converge on overall surprisingly similar diversity estimates (which is quite reassuring).

Minor, detailed comments:
- Abstract – in contrast to the main text, not that smoothly written. Eg., some repetitions at the beginning; statements that seem conflicting at the end – on one hand number of paralogs is overestimated in haploid genome, on the other- MHC alleles from amplicons are missing. Of course it can happen (and is explained in the text) but upon first read it is not entirely clear.

- l. 103-108 – Too long a sentence, hard to follow and understand.

- l. 227-228 – “Amplicon alleles mapping with mismatches to several MHC gene copies were not considered.” – what where these? Alleles for which loci were not present in the genome assembly? Or were too divergent within locus to be correctly assigned? Or were some erroneous sequences (e.g., chimeras?).

- paragraph starting in line 230 – “Coding sequence similarity for MHC-IIB haplotypes” – here the whole concept of testing whether “the scaffolds Aaru-DAB*554 (maternally inherited) and Aaru-DAB*357 (paternally inherited) represent the two parental haplotypes …” comes out of the blue – as it seems (from this sentence) that it is already somehow known which comes from which parent – rather than being discovered during the course of this study.

- l. 249-250 – “alleles within a gene copy” – is it to mean alleles from the same locus? As is phrasing is a bit convoluted.

- l. 258 – represents, should be in plural (represent)?

- l. 268 – is 6443 kb correct, or should be “bp”? Is this average intergenic distance in the genome, or between tandemly duplicated MHC-IIB genes?

-l. 351 – extra “in”: “in more in detail”

-l. 457-461 – have the assembled genome been checked for polymorphisms that could cause poor primer annealing in case of the annotated loci missed with amplicons, or is it just a speculation?

Fig 2. If Aaru-DAB*554 and 357 scaffolds represent maternal and paternal haplotypes of the same genomic region, respectively, how the first gene (#1) can be unequivocally maternal for DAB*554 and present in both parents (indicated by turquoise color) in DAB*357?

·

Basic reporting

1. The use of English language at a technical level is clear.

2. The article has the exact and sufficient references that support your research.

3. The structure of the article is well developed. Figures and tables match the text and are relevant.

4. The results and the general objective are coherent.

Experimental design

The experimental design meets the requirements of the journal. The methods are well described and developed, are precise and clear.

Validity of the findings

The results are relevant for future research in the area. The data in the article is robust and sufficient for your understanding. The conclusions are correct to the article. The statistical data is consistent with what was expected.

Additional comments

I have to congratulate the authors of the article for their excellent work. I have no objection to your publication. From my point of view no changes are needed.

---

## Round 0.2 · accepted · Accept

Dear authors,

It is my pleasure to inform you that the manuscript has been accepted, as both reviewers have expressed their satisfaction with the revisions made. I would like to express my gratitude for your diligent work in addressing the comments and making necessary corrections.

Thank you for your dedication and commitment to this manuscript.

Best regards,

Armando Sunny

Reviewer 1 ·

Basic reporting

No more comments

Experimental design

No more comments

Validity of the findings

No more comments

Additional comments

I thank the authors for taking my comments into account and amending the manuscript. I believe it can be now published in PeerJ.